# Establishing an Extracorporeal Cardiopulmonary Resuscitation Program

**DOI:** 10.3390/medicina60121979

**Published:** 2024-12-02

**Authors:** Pietro Bertini, Fabio Sangalli, Paolo Meani, Alberto Marabotti, Antonio Rubino, Sabino Scolletta, Valentina Ajello, Tommaso Aloisio, Massimo Baiocchi, Fabrizio Monaco, Marco Ranucci, Cristina Santonocito, Simona Silvetti, Filippo Sanfilippo, Gianluca Paternoster

**Affiliations:** 1Department of Anesthesia and Intensive Care Medicine, Casa di Cura Privata San Rossore, 56122 Pisa, Italy; pietro.bertini@gmail.com; 2Department of Anaesthesia and Intensive Care, ASST Valtellina e Alto Lario, University of Milano-Bicocca, 23020 Sondrio, Italy; 3Department of Cardiothoracic Surgery, Heart and Vascular Centre, Maastricht University Medical Centre, 6229 HX Maastricht, The Netherlands; 4Intensive Care Unit and Regional, ECMO Referral Centre, Azienda Ospedaliero, Universitaria Careggi, 50134 Florence, Italy; albertomarabotti@gmail.com; 5Royal Papworth Hospital NHS Foundation Trust, Cambridge CB2 0AY, UK; 6Department of Medical Science, Surgery and Neurosciences, Trauma Anesthesia and Intensive Care Unit, University Hospital of Siena, 53100 Siena, Italy; sabino.scolletta@dbm.unisi.it; 7Department of Cardiac Anesthesia, University of Tor Vergata, 00133 Rome, Italy; 8Department of Cardiothoracic and Vascular Anesthesia and Intensive Care Unit (ICU), Istituto di Ricovero e Cura a Carattere Scientifico (IRCCS) Policlinico San Donato, 20097 Milan, Italymarco.ranucci@grupposandonato.it (M.R.); 9Cardio-Thoracic and Vascular Anesthesia and Intensive Care Unit, IRCCS Azienda Ospedaliero-Universitaria di Bologna, 40126 Bologna, Italy; dr.massimo.baiocchi@gmail.com; 10Department of Anesthesia and Intensive Care, IRCCS San Raffaele Scientific Institute, 20132 Milan, Italy; monaco.fabrizio@hsr.it; 11Department of Anaesthesia and Intensive Care, Azienda Ospedaliero Universitaria Policlinico-San Marco, Site “Policlinico G. Rodolico”, 95123 Catania, Italy; cristina.santonocito@gmail.com; 12Department of Cardiac Anesthesia and Intensive Care, Ospedale Policlinico San Martino IRCCS, IRCCS Cardiovascular Network, 16132 Genova, Italy; 13Department of Anesthesia and Intensive Care, University Hospital Policlinico G. Rodolico-San Marco, 95123 Catania, Italy; filipposanfi@yahoo.it; 14Department of Surgery and Medical-Surgical Specialties, University of Catania, 95123 Catania, Italy; 15Department of Health Science, Anesthesia and ICU, School of Medicine, University of Basilicata San Carlo Hospital, 85100 Potenza, Italy

**Keywords:** ECPR, critical care, ECMO, cardiac arrest, ECLS

## Abstract

Extracorporeal cardiopulmonary resuscitation (ECPR) is a complex, life-saving procedure that uses mechanical support for patients with refractory cardiac arrest, representing the pinnacle of extracorporeal membrane oxygenation (ECMO) applications. Effective ECPR requires precise patient selection, rapid mobilization of a multidisciplinary team, and skilled cannulation techniques. Establishing a program necessitates a cohesive ECMO system that promotes interdisciplinary collaboration, which is essential for managing acute cardiogenic shock and severe pulmonary failure. ECPR is suited for selected patients, emphasizing the need to optimize every step of cardiac arrest management—from public education to advanced post-resuscitation care. The flexibility of ECMO teams allows them to manage various emergencies such as cardiogenic shock, massive pulmonary embolism, and severe asthma, showcasing the program’s adaptability. Launching an ECPR program involves addressing logistical, financial, and organizational challenges. This includes gaining administrative approval, assembling a diverse team, and crafting detailed protocols and training regimens. The development process entails organizing teams, refining protocols, and training extensively to ensure operational readiness. A systematic approach to building an ECPR program involves establishing a team, defining patient selection criteria, and evaluating caseloads. Critical elements like patient transport protocols and anticoagulation management are vital for the program’s success. In conclusion, initiating an ECPR program demands thorough planning, collaborative effort across specialties, and ongoing evaluation to improve outcomes in critical cardiac emergencies. This guide offers practical insights to support institutions in navigating the complexities of ECPR program development and maintenance.

## 1. Introduction

Extracorporeal cardiopulmonary resuscitation (ECPR) embodies the swift institution of mechanical cardiorespiratory aid for patients in life-threatening conditions, constituting the most urgent and technically demanding ECMO application. The urgency adds intricacy to patient selection, team deployment, multidisciplinary coordination, and cannulation techniques. ECPR implementation necessitates the comprehensive organization of ECMO systems and depends heavily on cross-disciplinary and cross-professional teamwork. These well-structured systems are essential for timely mechanical support during acute cardiogenic shock and severe pulmonary failure. For instance, the ECMO team can be mobilized to manage cases of cardiogenic shock in a cardiac catheterization lab, massive pulmonary embolism in the emergency department, or V-V ECMO for intractable asthma in the intensive care unit. Furthermore, it can be deployed for ECPR, regardless of where it is required.

Out-of-hospital cardiac arrests (OHCAs) affect approximately 350,000 individuals annually in the US, but only a meager 8.4% survive with favorable neurological outcomes [1]. In contrast, the EuReCa TWO [2] European study reported an average survival to hospital discharge of 8%, despite widespread variability in bystander CPR rates and outcomes across different countries. A correctly implemented extracorporeal cardiopulmonary resuscitation (ECPR) program can significantly improve these disheartening survival rates while attempting to preserve neurological function. 

The American Heart Association’s 2023 update suggests that ECPR can be used as a rescue strategy for certain patients when traditional CPR methods fail [3].

Establishing an ECPR program is a monumental and intimidating endeavor for any institution. This initiative can pose a significant challenge to any organization’s system and possibly uncover some existing deficiencies [4].

Every institution has unique challenges to overcome when creating an ECPR program. Peripheral non-academic hospitals without ECMO programs and large academic institutions with many physicians eager to contribute to the project face their unique, demanding tasks. Despite the challenges, these institutions can use these circumstances step by step to their advantage in developing their ECPR programs [5].

This manuscript aims to guide institutions in establishing an ECPR program, specifically addressing the current lack of structured, step-by-step methodologies for effective implementation. It seeks to bridge gaps in the literature regarding systematic approaches, cross-disciplinary integration, and long-term program sustainability in ECPR applications.

## 2. ECPR Program: Step by Step

The following steps provide a guideline for establishing an ECPR program based on the experience of most institutions. These steps can be categorized broadly as follows:Preliminary institutional approval and shared multidisciplinary intents;Clearly defining the ECLS criteria;Arranging a team of involved participants;Constructing fundamental systems and procedures;Conducting training sessions;Carrying out and evaluating the system.

In reality, many tasks within these categories coincide, but addressing them sequentially benefits many institutions. For example, initiating system development without the input and commitment of key stakeholders could lead to complications. Conversely, some degree of system development must precede team training and rehearsal.

For any healthcare initiative to succeed long-term, it must be adaptable and continually improving. Thus, a successful program should anticipate revisiting these stages of system planning as new challenges or opportunities for growth arise.

Given the variability in resources, patient demographics, and clinical settings across institutions, we recommend that each institution develop a tailored decision-making framework for assessing ECPR candidacy. This approach allows institutions to consider specific environmental and clinical factors, such as resource availability, team expertise, patient population characteristics, and institutional priorities, which may influence the optimal selection criteria for ECPR candidates.

This guide offers practical insights to support institutions in navigating the complexities of ECPR program development and maintenance.

## 3. Preliminary Institutional Approval and Shared Multidisciplinary Intents

As one contemplates initiating an ECPR program, two preliminary steps are vital. The proposal must first win the approval of the hospital management, considering both the costs and potential benefits. Without their support, the program cannot be successfully launched. Following this, it is crucial to confirm that a diverse team of professionals keen to see this project come to fruition exists. It needs a unified effort across various disciplines; if such support cannot be secured, serious consideration should be given to halting the program’s initiation at this stage.

Once these initial steps are completed, the focus can shift toward team organization, developing ECMO systems and protocols, and establishing the ongoing team training processes before the program can be activated. A comprehensive written plan is crucial to advance the process and keep all stakeholders updated [6].

## 4. Clearly Defining the ECLS Criteria

Effective Extracorporeal Life Support (ECLS) criteria are pivotal in optimizing patient outcomes by ensuring that the intervention is reserved for those most likely to benefit. Recent analyses from multi-regional Australian studies and specific case evaluations in Milan, Italy, offer profound insights into refining these criteria for enhanced clinical decision-making.

The study by Diehl et al. provides an insightful analysis of the impact of different ECLS (Extracorporeal Life Support) selection criteria on patient outcomes during extracorporeal cardiopulmonary resuscitation (ECPR) [7]. The research underscores a fundamental debate between employing restrictive versus liberal criteria in determining eligibility for ECLS.

The restrictive criteria delineated in the study were defined by stringent parameters, which included the following:Age: Eligibility only for patients under the age of 70.No-flow time: The time from cardiac arrest to the initiation of CPR must be under 12 min.Low-flow time: The duration from CPR commencement to ECLS initiation must be less than 60 min.End-tidal CO_2_: A measurement greater than 10 mmHg after 20 min of ongoing CPR.Shockable presenting rhythm.

Implementing these restrictive criteria meant that a smaller subset of all cardiac arrest patients was considered eligible for ECPR. However, this selectivity was associated with higher survival rates among those treated, suggesting that stringent criteria may help identify those most likely to benefit from this invasive intervention. The limited eligibility implies a focus on resource optimization, targeting the intervention to those with the highest likelihood of positive outcomes and thus potentially improving the cost-effectiveness and efficiency of healthcare delivery in emergency cardiac care.

Conversely, the liberal criteria expanded the eligibility to a broader patient demographic. This approach reduces the stringency of the parameters used to assess ECPR eligibility, thereby increasing the number of patients who qualify for the procedure. While this inclusivity ensures that more patients receive a potentially life-saving intervention, the study noted a consequential decrease in overall survival percentages.

The liberal approach operates under the principle that more lives can be saved by broadening the criteria, albeit with a possible reduction in the success rate per individual case. This method may be particularly justified in scenarios where the capacity for ECLS is underutilized or where healthcare providers aim to extend potential benefits to a broader group, reflecting a more equitable resource distribution but accepting the trade-off of potentially lower efficiency.

The findings by Diehl et al. highlight a critical balance that must be struck between maximizing the survival rates through selective criteria and broadening the availability of life-saving technology to a larger group [7]. The decision between restrictive and liberal criteria depends on the specific patient population and institutional capabilities as well as broader ethical, economic, and clinical considerations that influence policy decisions in emergency medical services. The study illustrates the complex interplay between clinical efficacy and resource allocation in deploying advanced medical interventions like ECLS.

### Influence of Initial Conditions

Adding to the complexity, Scquizzato et al. observed that specific initial conditions such as shockable rhythms and shorter low-flow times, significantly impacted the success rates of ECLS in out-of-hospital cardiac arrests (OHCAs) [8]. When the initial rhythm was shockable, and the low-flow duration was less than 60 min, survival and favorable neurological outcomes notably increased. 

## 5. Financial Costs Related to ECPR Program

Establishing an ECPR program involves several financial considerations that can significantly impact its success and sustainability [9,10]. Firstly, initial startup costs are a crucial factor. These may include purchasing advanced medical equipment such as ECMO (extracorporeal membrane oxygenation) machines as well as other supportive devices needed for monitoring and care during the procedure. Next, the program requires an investment in personnel training and development. Skilled staff, including critical care physicians, nurses, and paramedics, must be trained to operate highly specialized equipment and to understand the specific protocols of an ECPR program. This training can lead to higher labor costs initially. Operational costs also need to be considered. This includes day-to-day expenses such as the maintenance of equipment, consumables for procedures, and potential reimbursements for the program, which may vary depending on the healthcare system and insurance providers involved. Lastly, establishing partnerships with local hospitals and emergency services is vital for resource sharing and support. Overall, significant training, skill, equipment, and related expense is required to offer an effective ECPR program. Three main studies conducted in North America, Australia, and Japan showed that ECPR had reasonable costs for advanced medical interventions [11,12,13]. These costs ranged from $16,000 to $52,000/quality-adjusted life-years [14]. Overall, careful financial planning and continuous evaluation are essential for the sustainability of an ECPR program, ensuring that it delivers high-quality care effectively.

## 6. Arranging a Team of Involved Participants

### 6.1. ECPR Team Organization

Early/acute management. To design and implement a successful ECPR system, collaboration among different medical specialties and allied health professions is essential and should be emphasized. An example of an ECMO team organization is illustrated in Table 1. By focusing on a multi-professional and multidisciplinary approach, one will be better equipped to navigate each center’s unique challenges. To illustrate the spectrum of medical professionals who need to be part of the team, let us walk through a typical OHCA ECPR case.

In the event of sudden cardiac arrest, the emergency medical services (EMS) are activated, and prompt bystander CPR minimizes the time without blood flow. A 911/112 operator trained to screen for potential ECPR eligibility identifies the patient and relays this to the responding paramedics. The EMS providers then offer optimal, quick, initial ACLS, transitioning from a “stay and play” mentality to a brief ACLS algorithm. This involves rapid airway management and a mechanical CPR device and ensuring that there are no ECPR exclusion criteria. After this, a “scoop and run” transport plan is initiated as soon as possible to transfer the patient to and ECPR program facility to limit the low-flow period.

The ECPR team and the emergency department (ED) are activated via a standardized ECPR alert, leading to a well-coordinated “code” in the ED that utilizes role-based ACLS, a secondary ECPR criteria screen, and preparation for ECMO cannulation. The ECPR team performs intra-arrest cannulation under transesophageal echocardiography and/or fluoroscopy guidance provided by a trained EM or ECPR team member. Immediate post-cannulation issues may involve rapid fluoroscopy or a CT scan and interpretation, necessitating direct involvement from radiologists [15].

Next, the patient is transported to the cardiac catheterization lab (CCL), where the interventional cardiologist performs a stent placement with potential left heart decompression [16]. At this moment, the placement of lower extremity reperfusion cannulas should be considered [17].

Late/sub-acute management. Daily intensive care challenges are managed by a multidisciplinary team, including nurses, respiratory therapists, radiologists, pharmacists, clinical dietitians, physical therapists, and auxiliary staff. Complication management necessitates the collaboration of general surgery for operative care, infectious disease specialists for infection management, and the transfusion medicine team for blood product usage. Proper left ventricular unloading is critical, and cardiologists collaborate closely with the care team to optimize this aspect of patient support during cardiac recovery. Neurologists are also involved in neuroprognostication to assess and manage potential neurological outcomes [18]. Critical care physicians guide the weaning from mechanical support with cardiology expertise. Specialized surgical or intravascular devices can be used to remove arterial cannulas, ensuring safe and effective decannulation [19].

As mentioned earlier, all groups must be involved in creating the ECPR program. Even considering different institutional cultures, a successful ECPR program demands a bold and extensive multidisciplinary team effort to achieve optimal outcomes. Once these two foundational steps are secured, it is time to proceed with the organization of any ECPR program [15].

### 6.2. Building an ECPR Team

Forming an initial resuscitation committee with all relevant stakeholders is crucial and should occur well before the ECPR program goes live. Physician stakeholders include specialists from emergency medicine, intensive care, surgical fields (cardiothoracic, vascular, and general), cardiology, anesthesiology, radiology, and palliative care. Close collaboration between physicians, nurses, ECMO specialists, pharmacists, and bedside therapists is also necessary for the success of the ECPR team. Early on, the team must establish the ECPR mission and timeline. The chance for this mission to solve previous multidisciplinary challenges should not be overlooked. For instance, at the University of New Mexico, establishing an ECPR system facilitated advances in the resuscitative transesophageal echocardiography (TEE) program [20], improved role-based ACLS delivery, and fostered greater collaboration between ICU and ED nursing.

As one initiates early meetings with stakeholder groups, their specific goals, challenges, and concerns should be anticipated. Time to meet with local EMS leaders should be allocated. The ECPR center should understand and anticipate potential political challenges related to transporting and potentially redirecting suitable candidates to the ECPR center. An early step in the program involves inviting EMS leaders to educational presentations about the in-hospital ECPR program planned before the actual deployment of ECPRs for OHCAs. This engagement ignites enthusiasm among EMS leadership, leading them to work with 911/112 dispatchers and acquire automated CPR devices for performing CPR while transporting potential ECMO patients.

By anticipating the concerns from the cardiology team about cardiac catheterization lab (CCL) cases with a high risk of severe anoxic injury, improvements to our response for cardiogenic shock cases in the CCL can be made. After successfully applying ECPR to CCL patients, interventional cardiologists should be more eager to handle such cases from the ED. Similarly, potential conflicts over cannulation can be resolved through early, open discussions; a willingness to train together; and the intensivists’ readiness to shoulder most of the cannulation responsibility.

#### 6.2.1. Cannulation Team

One of the early decisions for creating an ECPR program was determining the size and composition of a cannulation team. ECPR cannulation can have potential complications, which can be minimized with repetition and volume of cannulation experience. If the team includes too many “cannulators”, no single physician receives significant exposure to the process, which could compromise their expertise in troubleshooting problems. Conversely, the team must be large enough to ensure adequate service coverage and not overtax a smaller group. Safe cannulation has been demonstrated across various specialties, which means the specialties of the cannulation team are less critical than the training and quality assurance system designed to optimize outcomes.

Training of “cannulators” can occur at national courses or home institutions through mentorship from previously trained physicians, such as cardiac/vascular surgeons. It is essential to establish and nurture relationships with institutional specialists for backup during the early stages of the program. A standardized cannulation privilege, set within the hospital, ensures that those wishing to cannulate have met the training requirements and have the appropriate knowledge of the processes and pitfalls. A stepwise progression, starting with establishing proficiency in basic vascular ultrasound and teaching wire manipulation techniques in elective cases, is advisable. This should be followed by supervised non-emergency cannulations and, ultimately, ECPR cases.

#### 6.2.2. Management Team

While many physicians may wish to provide cannulation services and initial resuscitative management, developing a strong ECMO management team for the post-cannulation period is essential. Cannulation takes minutes, while ECMO management can last days to weeks. We recommend that physicians tasked with the care of the ECMO patient, at a minimum, attend an ECMO management course offered by organizations like the Extracorporeal Life Support Organization (ELSO) [21].

Every ECMO program requires a qualified ECMO specialist support team. Several models exist for obtaining this expertise at the bedside: (1) utilizing outsourced ECMO specialists from a private perfusionist company; (2) expanding internally trained specialists from the cardiac surgery perfusionist service to provide bedside ECMO care; or (3) broadening the scope and training of ICU nurses, ED nurses, and respiratory therapists to include ECMO specialization.

Each model has its benefits and drawbacks, often impacted by local conditions. For example, utilizing internal ECMO specialists from the pool of ICU nurses and respiratory therapy staff can lead to staff shortages in the ICU. On the other hand, a private company specialist team may have contractual limitations. The chosen model should result in a close partnership in the ECMO process and day-to-day management. A common thread among the various groups is the leadership of a dedicated ECMO coordinator who oversees the ECMO specialist group functions as a bedside clinical resource, educator, and quality assurance leader.

#### 6.2.3. Patient Selection

Deciding which patients to treat is crucial for the initial stakeholder group. Developing inclusion/exclusion criteria that align with institutional capabilities is also crucial. 

There are approximately 290,000 in-hospital cardiac arrests each year in the US, with a mean age of 66. This population will provide some of the best candidates for an ECPR program. Ideal criteria include witnessed arrest, early application of essential life support, an initial shockable rhythm, and a known past medical history that excludes patients with end-stage liver, lung, or kidney disease or metastatic malignancy.

Also, the duration of CPR prior to ECMO is prognostic for survival [22]. Patients who arrest in the hospital have the best potential to be cannulated for ECMO. As in-hospital cardiac arrest (IHCA) patients are often monitored and receive the rapid institution of quality CPR, they meet the established goal of very short no-flow (no CPR) and low-flow (CPR only) time prior to V-A ECMO. A low-flow time of less than 60 min has been shown to improve survival rates significantly.

A specific cause of hemodynamic instability that may prove beneficial for a new program is massive pulmonary embolism, which has a mortality rate as high as 65%. If ECPR can be applied immediately following a cardiac arrest, the mortality rate can be significantly improved [23,24].

#### 6.2.4. Caseload Evaluation

ECPR patients impact multiple clinical service lines and units, so predicting and preparing for the increased caseload is an essential early step. Once the inclusion/exclusion criteria have been chosen, one should be able to approximate the potential target caseload from previous years’ cardiac arrest data from the ED, the CCL, and the total number of massive PE patients.

To optimize resource utilization and staffing, consider many variables, including off-hours resource availability, physician training, and ECMO console availability, depending on seasonal surges, pediatric needs, or post-cardiotomy ECMO volume at any institution. Local market saturation may also affect these calculations. If multiple hospitals in a small city offer ECPR, this may dilute the potential total volume of cases.

Caseload estimates facilitate the approximation of costs, including the quantity of ECMO consoles and disposables. Understanding these numbers and how they affect each stakeholder is crucial to the success of a program.

#### 6.2.5. Newborn ECPR Program: Early Challenges and Troubleshooting

Implementing an ECPR program poses several troubleshooting challenges during its first year. First of all, the most significant challenge is ensuring proper training and competency among the healthcare staff. ECPR requires specialized skills, and if the staff are inadequately trained, it can lead to improper device use or patient management, dramatically jeopardizing patient outcomes. Logistical issues also frequently arise. These can include difficulties in obtaining timely access to necessary equipment and ensuring that supplies are adequately stocked. Another common challenge is patient selection. Identifying which patients are appropriate candidates for ECPR can be complex, requiring clear criteria and consensus among the team. Inconsistent decision-making can lead to missed opportunities for eligible patients or inappropriate interventions [25]. Finally, capturing information about events—including hazardous conditions, near misses, adverse events, and sentinel events—helps an organization learn and improve continuously while creating safer care for patients and safer conditions for the staff. In this light, ongoing data collection and analysis are needed to evaluate program efficacy, which requires dedicated resources and commitment. In addition, a risk management training plan should be established, including audits and training that should be provided annually and when changes are made to the system. Addressing these challenges proactively can enhance the program’s success and improve patient outcomes.

## 7. Constructing Fundamental Systems and Procedures

Establishing an ECPR (extracorporeal cardiopulmonary resuscitation) team involves educating the members about the role and benefits of ECMO (extracorporeal membrane oxygenation) and designing critical systems and protocols for various situations. This should be tailored to the institution’s unique resources and settings while maintaining team consistency. Part of the process involves standardizing the team’s response to critical patients. This could involve collaborating with the existing “shock team” or creating a new process altogether. A rapid response is crucial for enhanced patient outcomes. Another key aspect is the optimal delivery of Advanced Cardiovascular Life Support (ACLS) through a systematic and role-based approach coordinated by a resuscitation leader.

The location for ECPR should be strategically chosen based on the specific needs of the procedure and the available institutional resources. Options include the emergency department (ED), which offers immediate access but may lack specialized equipment; the catheterization laboratory (cath lab), which is equipped for complex interventions but may not be immediately available; and the hybrid operating room (OR), which combines the readiness of the ED with the equipment of the cath lab but requires significant logistical coordination. Advantages and disadvantages of the common places where ECPR is carried out are shown in Table 2.

Patient transport can be complex and high-risk, requiring a standardized approach and checklist to ensure that ECMO support is maintained throughout. Benchmarking the program’s complications against established figures like those in the Extracorporeal Life Support Organization (ELSO) database is crucial, triggering reassessment whenever a deviation occurs. Quick, appropriate adjustments should be made to procedures if complications trend upwards. The anticoagulation protocol is critical to managing the balance between thrombotic and hemorrhagic complications for different ECPR patients. This includes managing specific challenges such as massive pulmonary embolism, dual antiplatelet medication administration during cardiac catheterization, and CPR-induced trauma. Regarding coronary angiography, collaboration with interventional cardiologists is crucial for patient selection and timing of angiography and Percutaneous Coronary Intervention (PCI) in ECPR and cardiogenic shock patients. It is beneficial to conduct a full-body CT scan before angiography to limit trips to the cardiac catheterization lab (CCL) if the cause of arrest is noncardiac.

The ECPR service should undergo regular reviews, focusing on prearranged indications, complications during cannulation and ICU management, and outcomes. In addition to survival rates, we suggest tracking other benchmarks such as neurological outcomes, procedural efficiency (e.g., time to cannulation), and patient quality of life post-recovery. These metrics offer a more comprehensive view of program success and can highlight areas for targeted improvement.

To ensure continuous improvement, we recommend using structured evaluation tools, such as checklists or scoring systems, to assess team performance during simulations. Key metrics might include response times, adherence to protocols, and inter-team communication efficiency. Regular reviews of these metrics allow for tracking improvements and addressing identified challenges.

The implementation of safe and skilled cannulation, standardization of ECMO equipment carts, and preparation for possible complications such as extremity ischemia or left heart decompression are crucial (Figure 1). It is essential to incorporate best practices into protocols while remaining flexible in areas of uncertainty, as each patient’s case is unique. Standardization improves outcomes and reduces the chance of error by ensuring that every team member knows what to expect and how to respond.

Anticoagulation protocols, coronary angiography, and quality assurance are vital elements of ECPR program implementation. Regular reviews focusing on adherence to indications, complications, and outcomes are vital, with survival being only part of the measure of success. Additionally, the integration of Donation After Circulatory Death (DCD) within ECPR protocols offers a critical extension to enhance the program’s utility by potentially increasing the donor pool, which is essential given the profound organ shortage [26]. This approach respects both the donor’s dignity and the recipient’s medical needs, providing a valuable source for transplantation even when ECPR does not result in patient survival. Proficient procedures in cannulation, extremity reperfusion, left heart decompression, and preparing for unanticipated operations are integral to the ECPR program. Practitioners should be well trained to handle cannulation under pressure and troubleshoot any issues. Standardized ECMO equipment carts organized in a manner that mirrors the cannulation process should be consistently stocked and strategically located. Moreover, general surgeons should be comfortable operating on ECMO patients, given the potential for injuries necessitating an immediate laparotomy. Ultimately, the key lies in communication and planning to manage these complex situations.

## 8. Conducting Training Sessions

Once consensus on vital operational aspects is achieved and initial designs for various systems are created, multidisciplinary team training can commence. Education initiatives should include lectures, reference materials, or videos introducing ECMO systems to different groups, targeting every previously mentioned stakeholder. Feedback from these sessions can aid in continuously improving the initial design.

However, lectures alone are insufficient. Regular in situ simulations should form a significant component of ECPR system training and cover every process aspect, including ACLS, ECMO cannulation, patient imaging, transport, etc.

Daily bedside rounds can serve as an essential learning opportunity due to the complexity of ECPR patients and their management challenges. Routine morbidity and mortality reviews are crucial for ongoing education and quality control.

## 9. Carrying Out and Evaluating the System: Where to Go from Here

After many months of effort, the program organization and creation of a multidisciplinary team should be complete. All tasks from the initial logistical plan should be checked: hospital leadership support, regular team meetings, clear role understanding, and an organized stakeholder team with well-designed critical systems, protocols, and training.

Details such as selecting ECMO devices and cannulas, designing ECMO order sets in electronic medical record systems, and training coders to identify critical care and ECMO billing nuances, among others, must not be overlooked.

The next step is to announce the go-live date for the ECPR program with a hospital-wide launch event that promotes excitement and educates about the purpose of ECPR, its indications, and potential benefits to the hospital population.

In anticipation of the first patient, the cannulation team should frequently review the indications, exclusionary criteria, and the standardized cannulation process. The ECPR team should evaluate all potential activations closely, attend all cardiac arrests and potential resuscitations, and provide support for intra-arrest arterial and venous access when the patient is not a candidate for ECMO.

The success of the first patient will be a culmination of 10–12 months of labor and preparation. With these efforts, the ECPR program will be primed for success and ready to deliver the highest resuscitation level to the sickest patients.

## Figures and Tables

**Figure 1 medicina-60-01979-f001:**
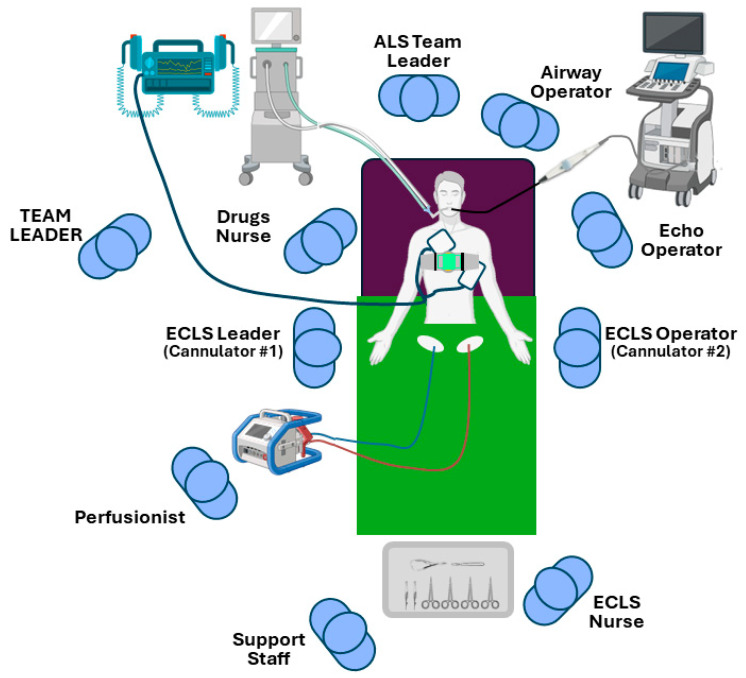
Schematic drawing of roles and procedures during an ECPR cannulation.

**Table 1 medicina-60-01979-t001:** Example of the professional roles and tasks involved in the early/acute management of patients arriving in an extracorporeal cardiopulmonary resuscitation (ECPR) setting, based on the ECPR team composition and responsibilities.

Team Member	Role	Primary Responsibilities
Consultant (leader)	ICU training, leader of CPR team	Direct the overall management of the procedure, ensure protocol adherence, make critical decisions during ECPR
Perfusionist	Pump operator, supervisor	Operate the ECMO pump, oversee its functionality throughout the procedure, assist in critical decision-making
Intensivist	Responsible for cannulation	Perform cannulation procedures, oversee patient stabilization and integration to ECPR system
Physician in training	General support	Assist in patient care under supervision, engage in various procedural tasks as needed
Specialized nurses [3]	Intensive care support	Monitor patient vitals, administer medications and support as needed, assist with mechanical ventilation
Anesthesiologist/intensive care specialist	Cardiothoracic specialist, responsible for TEE	Conduct transesophageal echocardiography to assess cardiac function and guide treatment during ECPR

**Table 2 medicina-60-01979-t002:** Comparison of different locations for conducting extracorporeal cardiopulmonary resuscitation (ECPR) based on the advantages and disadvantages of each location.

Location	Advantages	Disadvantages
Emergency department (ED)	Immediate access for rapid response; suitable for initial stabilization	May lack specialized equipment and space for complex interventions; potentially high traffic and variability in patient volume
Catheterization laboratory (Cath Lab)	Well equipped for complex interventions such as Percutaneous Coronary Intervention (PCI); dedicated environment	Not always immediately available; may be delayed access during off-hours or if already in use
Hybrid operating room (OR)	Combines readiness of the ED and equipment of the Cath Lab; versatile for various procedures	Requires significant logistical coordination; higher operational costs; may not be readily available in all institutions
Intensive care unit (ICU)	Familiar environment for ongoing patient care, facilitating post-ECPR management	May not have immediate access to specialized equipment like fluoroscopy; not always available for rapid deployment of ECPR

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
