# Peer review of "Establishing an Extracorporeal Cardiopulmonary Resuscitation Program"

_medicina, 2024, doi:10.3390/medicina60121979_

Round 1

Reviewer 1 Report

Comments and Suggestions for Authors

Dear Authors,

Thank you for submitting your manuscript titled “Establishing an Extracorporeal Cardiopulmonary Resuscitation Program” for consideration. I was pleased to receive it as a reviewer.

Your guide is a useful contribution, as it provides practical insights for institutions looking to develop their own ECPR capabilities. Strengths of your manuscript include its detailed step-by-step approach, clear organizational framework, and thorough consideration of multidisciplinary team requirements.

Here are my comments and suggestions for further improving the manuscript:

1. In the Introduction, could you clearly state the manuscript's aims? What gaps in the literature does your guide aim to address?

2. Consider including approximate timeline recommendations for each of the six main steps to help institutions better plan their program development.

3. Consider including a decision-making algorithm or flowchart to help teams rapidly assess ECPR candidacy in time-critical situations.

4. Could you provide guidance on how to evaluate and document team performance during simulations to track improvement over time?

5. Are there any benchmarks that you recommend for evaluating program success beyond survival rates?

6. Consider discussing the financial considerations involved in establishing an ECPR program (e.g., staffing costs, training expenses, reimbursement strategies)?

7. Consider adding a troubleshooting section addressing common challenges faced during the first year of program implementation.

8. Consider including an example protocol or flowchart to guide initial management decisions regarding anticoagulation.

9. Consider including guidance on establishing systematic methods for capturing and analyzing near-misses and adverse events to improve program safety.

10. Consider including sample checklists or standard operating procedures in an appendix for critical processes like team activation and cannulation preparation.

Thank you again for the opportunity to review your work.

Author Response

We thank the editor and the reviewers for their precious advices and the efforts to helping us to improve the manuscript’s quality.

My best regards,

Prof. Gianluca Paternoster,

corresponding author.

---------

Here are my comments and suggestions for further improving the manuscript:

1. In the Introduction, could you clearly state the manuscript's aims? What gaps in the literature does your guide aim to address?

We thank the reviewer for this important advice, the text now reads: "This manuscript aims to guide institutions in establishing an ECPR program, specifically addressing the current lack of structured, step-by-step methodologies for effective implementation. It seeks to bridge gaps in the literature regarding systematic approaches, cross-disciplinary integration, and long-term program sustainability in ECPR applications."

2. Consider including approximate timeline recommendations for each of the six main steps to help institutions better plan their program development.

We really appreciate the reviewer’s comment: The text now reads “Given the variability in resources, patient demographics, and clinical settings across institutions, we recommend that each institution develop a tailored decision-making framework for assessing ECPR candidacy. This approach allows institutions to consider specific environmental and clinical factors, such as resource availability, team expertise, patient population characteristics, and institutional priorities, which may influence the optimal selection criteria for ECPR candidates”.

3. Consider including a decision-making algorithm or flowchart to help teams rapidly assess ECPR candidacy in time-critical situations.

Many thanks indeed for this comment. Rather than applying a universal algorithm, we suggest a flexible framework that can be adapted according to local needs. Institutions might consider factors such as witnessed arrest, initial shockable rhythm, no-flow duration, and the presence of reversible causes, alongside institution-specific considerations such as staffing patterns and available technology. The development of such a framework should involve a multidisciplinary team to ensure that it aligns with the institution's overall mission and values.

The text now reads: “Given the variability in resources, patient demographics, and clinical settings across institutions, we recommend that each institution develop a tailored decision-making framework for assessing ECPR candidacy. This approach allows institutions to consider specific environmental and clinical factors, such as resource availability, team expertise, patient population characteristics, and institutional priorities, which may influence the optimal selection criteria for ECPR candidates”.

4. Could you provide guidance on how to evaluate and document team performance during simulations to track improvement over time?

Many thanks for this comment, the text now reads: "To ensure continuous improvement, we recommend using structured evaluation tools, such as checklists or scoring systems, to assess team performance during simulations. Key metrics might include response times, adherence to protocols, and inter-team communication efficiency. Regular review of these metrics allows for tracking improvements and addressing identified challenges."

5. Are there any benchmarks that you recommend for evaluating program success beyond survival rates?

Thank you for the comment, the text now reads: "In addition to survival rates, we suggest tracking other benchmarks such as neurological outcomes, procedural efficiency (e.g., time to cannulation), and patient quality of life post-recovery. These metrics offer a more comprehensive view of program success and can highlight areas for targeted improvement."

------------------------------------------

We thank the reviewer for their  time spent carefully reviewing the manuscript, and in their opinions regarding the science and presentation of the material.

6. Consider discussing the financial considerations involved in establishing an ECPR program (e.g., staffing costs, training expenses, reimbursement strategies)?

A specific paragraph has been added discussing the financial costs related to ECPR program.

7. Consider adding a troubleshooting section addressing common challenges faced during the first year of program implementation.

We fully agree with this comment.  A dedicated paragraph has been included (“ Newborn ECPR program: Eearly challenges and troubleshooting”)

8. Consider including an example protocol or flowchart to guide initial management decisions regarding anticoagulation.

We agree with the reviewer that an example flowchart to guide initial anticoagulation might be helpful and can improve our manuscript details. However, there are several medical specific topics which should be additionally addressed. Therefore, we would rather keep this manuscript focus on organizational models and structures, instead of treating specifically the medical management.

9. Consider including guidance on establishing systematic methods for capturing and analyzing near-misses and adverse events to improve program safety.

This topic has been included in the paragraph “Newborn ECPR program: Early challenges and troubleshooting”.

10. Consider including sample checklists or standard operating procedures in an appendix for critical processes like team activation and cannulation preparation.

Although the comment rises an interesting topic, team activation and cannulation preparation are extremely variable and therefore influenced by several local factors ( such as urban vs rural area, ECMO implantation location, hospital organization model, medical equipment and populations, i.e North America vs Europe).

Reviewer 2 Report

Comments and Suggestions for Authors

Thank you for the opportunity to review this manuscript submitted by the authors.

In my view, the proposed model of in-hospital ECMO implantation is already established in France. Therefore I have some key points that should be taken into consideration. 

  1. The fundamental question remains whether in-hospital ECMO implantation yields better outcomes compared to on-site implantation.
  2. Have the authors estimated the average time it takes for patients to reach the hospital? It would be important to assess how many patients who are eligible at the time of cardiac arrest remain eligible upon arrival.
  3. As you mentioned, patients with shockable rhythm have significantly great survival rate, thus this should be an eligibility criteria. 
  4. Have you researched the French model? Why is this program better than the French one?  

Author Response

Thank you for the opportunity to review this manuscript submitted by the authors.

In my view, the proposed model of in-hospital ECMO implantation is already established in France. Therefore I have some key points that should be taken into consideration. 

  1. The fundamental question remains whether in-hospital ECMO implantation yields better outcomes compared to on-site implantation.

Thank you for this observation. The point you underlined is indeed a critical consideration, but it remains challenging to evaluate due to a lack of robust studies that can provide definitive answers. While some evidence suggests that minimizing the time to ECMO initiation can improve outcomes, the balance between ensuring adequate resources and expertise in an in-hospital setting versus the potential time savings of on-site initiation remains an area that requires further research. This manuscript does not aim to resolve this debate but to provide practical guidance for those aiming to establish an in-hospital extracorporeal cardiopulmonary resuscitation (eCPR) program.

2. Have the authors estimated the average time it takes for patients to reach the hospital? It would be important to assess how many patients who are eligible at the time of cardiac arrest remain eligible upon arrival.

Thank you so much for your comments. The objective of this manuscript is to provide general guidance for those intending to initiate an eCPR program in their hospital. While the estimation of time from cardiac arrest to hospital arrival is indeed crucial for determining patient eligibility, it is highly dependent on the local context, including geographic factors, prehospital systems, and transportation infrastructure. As such, a universal estimate is not feasible within the scope of this document. However, we strongly encourage institutions to evaluate these parameters within their specific settings, as this data can critically inform program design and implementation.

3. As you mentioned, patients with shockable rhythm have significantly great survival rate, thus this should be an eligibility criteria. 

Thank you for emphasizing this point. We fully agree that the presence of a shockable rhythm is a significant predictor of better survival outcomes and should therefore be a key inclusion criterion. As indicated in the manuscript, our proposed eligibility criteria reflect this evidence, prioritizing patients with a shockable rhythm as candidates for eCPR. This aligns with existing literature and clinical practice guidelines, which consistently highlight the prognostic value of rhythm type in cardiac arrest scenarios.

4. Have you researched the French model? Why is this program better than the French one?  

The purpose of this manuscript is not to compare or promote one model over another but to propose a structured organizational framework for hospitals seeking to establish an eCPR program. We are indeed familiar with the French model, which is well-regarded for its success and efficiency. However, the aim of this handbook is to outline a practical, adaptable approach that can be tailored to the specific needs and resources of individual institutions. Each model, including the French one, has its unique strengths and limitations, and our intention is to empower hospitals to develop a system that works best within their local context rather than advocate for a single "better" solution.